# Centre-Negative: An Effective and Efficient Solution to Centre Bias in Visual Saliency Evaluation

**Author Name1**[*]   ABC@SAMPLE.COM  and  **Author Name2**   XYZ@SAMPLE.COM

**Editor:** Editor's name

## Abstract

Spatial bias is a long-standing problem in visual saliency detection, and various evaluation metrics have been proposed to address this issue. In this paper, we first review existing bias-specific saliency metrics, we group them into fixation-based and region-based and show that all of the metrics suffer from different drawbacks, especially when most fixations are closely distributed near the centre as is typical. To solve this problem, we study the essence of spatial bias in visual saliency. We show that the bias could be important in modelling saliency signal, thus ignoring or penalizing central regions cannot measure a saliency model comprehensively. The bias becomes problematic when comparing different saliency algorithms, a model with a strong central preference can capture most fixations. From this perspective, we propose a region-based metric-agnostic solution called Centre-Negative. The proposed approach can deliver three main advantages: a) our solution is not designed for a saliency metric specifically, Centre-Negative can be combined with any existing metrics to make use of their properties and simultaneously overcome the centre bias problem; b) our method is a region-based solution, so it can handle the situation where most fixations are densely distributed near the centre; c) our method can be applied simply and efficiently with a time complexity of $\mathcal{O}(1)$, and we show that the negative map created by Centre-Negative outperforms other solutions.

**Keywords:** Centre Bias, Saliency Detection, Evaluation Metric

## 1. Introduction

Visual saliency prediction methods have been used in various applications, including object detection (Zhu et al., 2012; Cai et al., 2019), video compression (Zhu and Xu, 2018; Zhu et al., 2018), and quality assessment (Jia and Zhang, 2018; Yang et al., 2019). However, building successful saliency predictions is still challenging due to the difficulty in evaluation. Existing metrics cannot measure saliency comprehensively, as each metric has its own drawback (Bylinskii et al., 2018; Riche et al., 2013). A common solution is to apply multiple metrics for comparison (Kümmerer et al.; Jiang et al., 2015), but only few are proposed specifically for the problem of centre bias.

Given a natural image, a saliency model can predict which regions could be more interesting to humans based on the stimuli of the image. However, photographers normally place the most interesting region near the centre of an image (Parkhurst et al., 2002; Parkhurst and Niebur, 2003; Tatler et al., 2005b). Thus, a saliency model with strong central fixation bias can achieve high performances on non bias-specific metrics (Bruce et al., 2015). And placing a 2D Gaussian map could outperform some well-designed saliency predictions

---

[*] with a note

due to the centre bias of the Human Visual System (HVS), see Sec. 2. To solve this problem, several bias-specific metrics have been proposed, including shuffled Area Under Curve (s-AUC) (Tatler et al., 2005a), Centre-Subtraction (Centre-Sub) (Bruce et al., 2016), Spatially Binned ROC (spROC) (Wloka and Tstotsos, 2016) and Farthest-Neighbour-AUC (FN-AUC) (Jia and Bruce, 2020). In this study, we group them into two categories: a) Fixation-based: s-AUC, Centre-Sub, FN-AUC; b) Region-based: spROC.

The categorization of the bias-specific metrics is defined based on how the method addresses centre bias. Fixation-based metrics make use of the fixations within the same dataset. For instance, s-AUC and FN-AUC draw negative points from the fixations of other samples, Centre-Sub subtracts the average fixation map of the whole dataset from each sample. This type of metric completely relies on fixation points collected, but fixation-based metrics may fail to measure saliency when most fixations are closely distributed near the central region. In contrast, the region-based metric, spROC, computes saliency properties based on different regions within an image. However, one main drawback of spROC is that it is difficult to choose a proper size for annulus, spROC may fail to address the centre bias problem when the number of fixations each bin receives are quite different. We can also choose the size based on the number of fixations each bin has, but it could be time consuming to iteratively search for the parameter for each image, and spROC will be fixation-based in this scenario. More importantly, all of the bias-specific solutions could fail in saliency evaluation when most fixations are near the centre, see Sec. 2.1. Furthermore, spROC, s-AUC and FN-AUC were proposed based on AUC properties, which tend to ignore false positives with small values (Bylinskii et al., 2018).

All of the bias-specific metrics attempt to mimic the centre bias by using the fixations. However, we raise a new question in this study, can we use a pre-defined spatial bias map instead of mimicking if the bias is known (e.g., centre bias)? We propose a new saliency metric to address the centre bias problem. In framing the position of this new metric, we emphasize that a new metric for a task should be model-agnostic, drawbacks of existing metrics should be studied and discussed in theory, and the newly proposed metric should be explained with regards to why it can overcome existing drawbacks resulting in improved evaluation. This is in contrast to proposing or choosing metrics based on model performance. For instance, Itti (Itti et al., 1998) et al.. achieve lower scores than ImageSig (Hou et al., 2012) (and they are both lower than the network-based systems (Cornia et al., 2016; Cornia et al., 2018)) on all of the MIT/Tübingen[1] metrics, however, the metrics measure saliency very differently. Bylinskii et al.(Bylinskii et al., 2018) studied the differences among saliency metrics to show which are most trustworthy, even though in many cases similar model rankings are seen.

We start with analyzing the effect of spatial bias in saliency prediction, we show that spatial signal is useful information in learning saliency stimuli, especially when visual cues are not discriminative enough. Thus, we should not blindly penalize or ignore the centre region. The spatial (centre) bias becomes problematic when evaluating and comparing models, see Sec. 3.1. Following this idea, we propose a new centre bias solution for model evaluation, denoted as Centre-Negative (Centre-Neg), which has following advantages: a) our proposed Centre-Neg is region-based, which can penalize centre bias regardless of fixa-

---

1. https://saliency.tuebingen.ai/results.html

tion distributions. b) Centre-Neg is a metric-agnostic solution in that any saliency metrics can be combined with our Centre-Neg to fairly measure saliency, e.g., Pearson Correlation Coefficient (CC) recommended in (Bylinskii et al., 2018); c) Centre-Neg is very simple and efficient, which can be applied with a time complexity of $\mathcal{O}(1)$ comparing to FN-AUC, $\mathcal{O}(n)$.

In our experiments, we validate the effectiveness of Centre-Neg, we show that our method can better penalize centre bias on all of the metrics used, SIMilarity (SIM), Kullback–Leibler Divergence (KLD), CC, Normalized Scanpath Saliency (NSS) and the AUC family. We also compare our method against existing bias-specific metrics, we qualitatively and quantitatively show that the negative map created by Centre-Neg can penalize centre bias more and hurt fixations less, comparing to the negative maps built by s-AUC and FN-AUC. Finally, we show the impact of Centre-Neg on various low-level ("early-vision") saliency models, the CB-map cannot outperform other designed saliency systems using our solution, see Sec. 4.

## 2. Problem Formulation

We begin by briefly discussing the reason behind centre bias and how existing bias-specific solutions address this spatial problem. The behaviour of the HVS is widely studied in cognitive science, two main reasons are behind centre bias: a) Firstly, observers initially tend to fixate at the centre of an image when free viewing; b) Secondly, photographers tend to place the most interesting (salient) objects near the centre region, thus observers are very likely pay more attention to the centre. As a result, fixation points collected are mainly distributed near the centre of an image, which causes the centre bias problem in saliency detection (Parkhurst et al., 2002; Parkhurst and Niebur, 2003; Tatler et al., 2005b). Because a centred Gaussian distribution can capture most of the fixations, outperforming well-designed saliency predictions (Judd et al., 2012; Kümmerer et al.) regardless of the visual stimuli within a photo. Several bias-specific metrics have been proposed to penalize this "spatial" bias, but each metric still cannot measure saliency accurately.

### 2.1. Study of Existing Solutions

This section elaborates how those metrics address centre bias and what drawbacks they suffer from, including s-AUC (Tatler et al., 2005a), FN-AUC (Jia and Bruce, 2020), spROC (Wloka and Tstotsos, 2016) and Centre-Sub (Bruce et al., 2016).

s-AUC is the most commonly used metric in saliency benchmarks. This metric is an AUC-based metric, which draws a negative set from the fixations of other images within the same dataset. The negative set can penalize the CB-map because it is also near the centre region. Bruce *et al.*(Bruce et al., 2016) proposed a ground-truth creation process, Centre-Sub, in which the average density ground-truth map could be subtracted, optionally, from each map to overcome the centre bias. Both solutions are based on the fixations of the whole dataset, because they consider that the spatial bias delivered by the overall distribution of fixations can be used to overcome centre bias. Obviously, this type of method could falsely measure saliency regardless of the location of the fixation, s-AUC will penalize and Centre-Sub will ignore the centre region blindly. When all of the fixations are closely distributed near the centre, as shown in Figs. 1a and 1b, there will be almost no fixations left for saliency evaluation using Centre-Sub, Fig. 1c. The negative map created by s-AUC, Fig. 1d will be very close to the central region (positives).

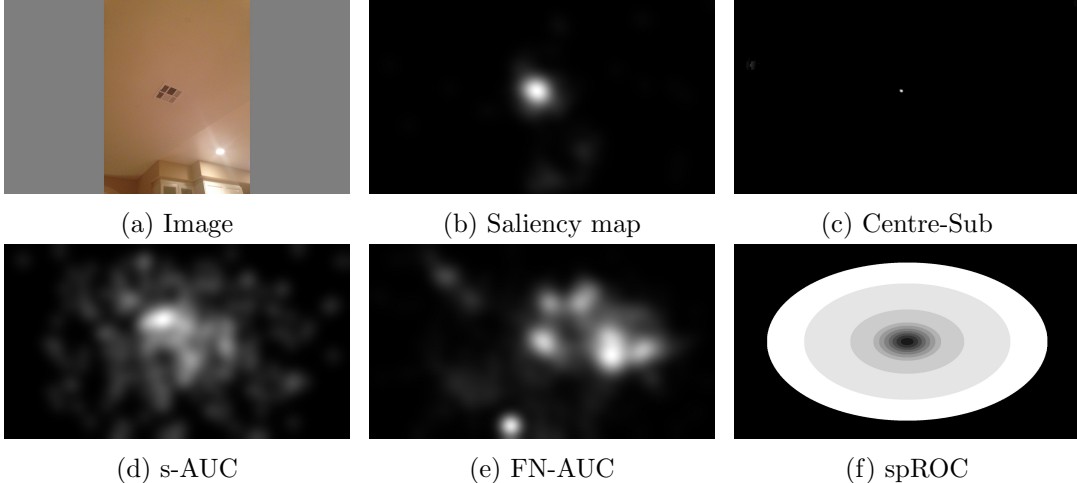

(a) Image      (b) Saliency map      (c) Centre-Sub

(d) s-AUC      (e) FN-AUC      (f) spROC

Figure 1: Examples of different bias-specific metrics when the fixations are closely distributed near the centre. (a) A natural image sample. (b) Ground-truth saliency map. (c) The subtracted saliency map created by Centre-Sub. (d) The negative set of s-AUC. (e) The negative set of FN-AUC. (f) The annulus of drawn by spROC.

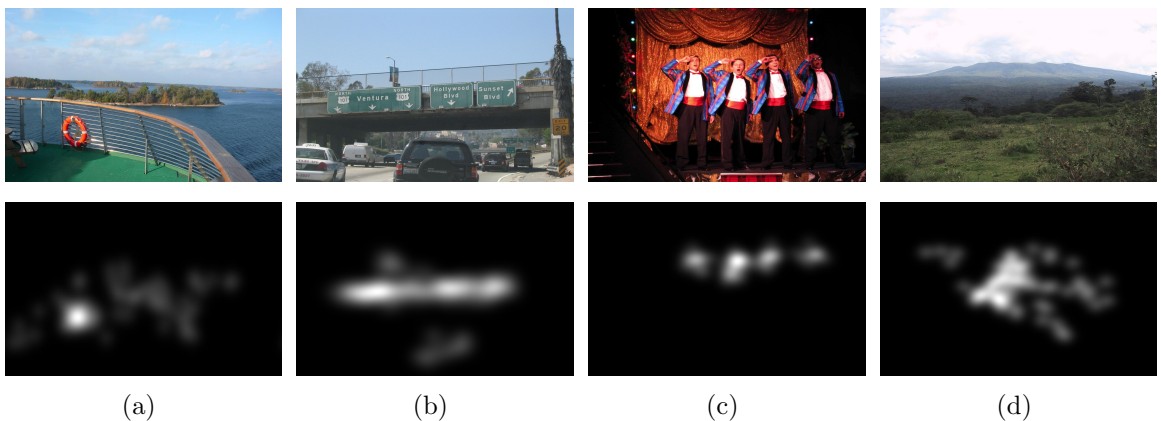

(a)      (b)      (c)      (d)

Figure 2: Examples of different salient regions. The top row shows sample images, and the bottom row shows their fixations. (a) A simple scenario in which there is only one prominent object. (b) and (c) Examples contain multiple objects which share the same object and low-level features, position information could be determinant. (d) There is no prominent object in the photo, the central region receives most of the fixations.

FN-AUC is a recently proposed metric, which also utilizes fixations from other images but aims at sampling a more directional negative set. However, FN-AUC is computationally expensive as it must iterate the whole dataset to sample a negative set for each image, with a time complexity of $\mathcal{O}(n)$. Furthermore, FN-AUC is also a fixation-based metric, which cannot solve the problem of the example shown in Fig. 1. FN-AUC will reduce to s-AUC when most of the fixations within a dataset are closely distributed within a specific (centre)

region. The sampled negative set could overlap with fixations, thus true positives could be penalized.

spROC was proposed to overcome the centre bias without using fixations from other images, defined as a region-based metric in this paper. spROC attempts to partition the entire image into a set of non-overlapping spatial annuli (bins). Then saliency will be measured separately within each bin and the performance gap among bins is an indicator to show spatial preference. However, it is very challenging to choose the size for each annulus. As suggested in (Wloka and Tstotsos, 2016), one can choose the annulus size based on the size of the input image, but this cannot guarantee that each annulus will receive the same amount of fixations. As shown in Fig. 1f, most of the fixations will fall into the inner most bins, but no fixations are in the outer bins for evaluation. Alternatively, one can set each annulus to receive an equal portion of the total fixations (spROC will become fixation-based). In this scenario, the CB-map can achieve similar results in each bin and no spatial biases can be detected. Moreover, it is time consuming to search for the optimal size for each bin. The requirement of spROC is higher than FN-AUC, because FN-AUC could fail when all the fixations within a dataset are closely distributed, while spROC may fail when it happens on one sample.

To summarize, the available solutions to centre bias have the following problems: a) they penalize or ignore the potential positive centre region; b) they can not build a meaningful negative map when most fixations distribute near the central region; c) the majority of available solutions are computed on AUC statistics, which omit false positives with small values (Bylinskii et al., 2018). Our proposed Centre-Neg solution can overcome these problems. We apply our solution in a metric-agnostic fashion, any saliency metrics can be applied on the output map of Centre-Neg to receive the advantages of this method, e.g., CC, NSS, AUC, SIM, and KLD. Our experiments show that the negative map created by Centre-Neg can better penalize the CB-map comparing with s-AUC and FN-AUC, see Sec. 4.3.

## 3. Methodology

### 3.1. Revisit the Impact of Spatial Bias

In designing a saliency prediction algorithm, the main question is: What makes a region salient? There are various hypotheses and study designs to answer this question, and some results are contradictory. An initial saliency study (Parkhurst et al., 2002) showed that stimulus-driven, bottom-up mechanisms determine attentional guidance under natural viewing, e.g., colour, intensity, and orientation (known as "early" features). Later, a contrasting hypothesis was raised by Einhäuser et al.(Einhäuser et al., 2008), in which subjects were shown an image and asked to recall the object's names and manually draw object locations that represent human fixations. The proposed top-down approach is based on more abstract visual features, i.e., objectness rather than low-level features of the previous study. Their study concluded that "early" saliency indirectly affects attention. Instead, "interesting" objects have a more considerable impact on learning saliency. However, later Borji et al.(Borji et al., 2013) showed that the object map (Einhäuser et al., 2008) achieves very similar results to the "early" feature-based saliency model (Itti et al., 1998), when compared with s-AUC metric.

Based on previous studies (Parkhurst et al., 2002; Einhäuser et al., 2008; Borji et al., 2013), we can conjecture that the subjects can easily recall (pay attention to) the object when the scene is relatively simple, as shown in Fig. 2a. The fixation points in this image show more concentration on the lifebuoy. A model capable of detecting regions with a high objectness score can perform well in these scenarios. One may argue that low-level features like colour function well for this image, which is a fair statement. However, if multiple regions contain similar objects and low-level features, the position information could determine saliency. As shown in Fig. 2b, the road signs at the centre draw most of the attention. In contrast, the road sign at the back receives almost no fixations. Similarly, in Fig. 2c, the middle person (second to the left) receives more attention due to his central position. If there are no prominent objects shown in the photo, Fig. 2d, the centre region still receives most of the attention. Einhäuser *et al.*(Einhäuser et al., 2008) also validated the correlation between position and saliency.

Thus, we can conclude that the position bias could be useful in modelling the saliency signal, especially when the high-level object and the low-level "early" features are not discriminative. The position bias becomes problematic only when we want to evaluate different saliency systems as discussed in Sec. 2. The centre region could be our friend, which should not be penalized or ignored. Instead, the CB-map is our foe when evaluating saliency.

## 3.2. Centre-Negative

We proposed Centre-Neg to address the drawbacks of the available solutions to the centre-bias problem in saliency evaluation. Let $\mathbf{X} \in \mathcal{R}^{h \times w \times 3}$ and $\mathbf{Y} \in \mathcal{R}^{h \times w}$ denote the input image and its ground-truth saliency map respectively,where $h$ and $w$ are the height and width of the image. Let $\mathbf{C}$ denote the CB-map. The values of $\mathbf{Y}$ and $\mathbf{C}$ are normalized into the range of $[0, 1]$.

Fig. 3 shows the flowchart of the proposed method. The density map, $\mathbf{Y}$, is enhanced by setting a threshold $\epsilon$. This parameter $\epsilon$ controls the extent of the enhancement. Smaller $\epsilon$ values will build a more enhanced ground-truth, thus the negative set will be farther to the fixations. (Our study applies a fixed value of $\epsilon = 0.1$). Each value of $\mathbf{Y}$ will be set to one if the value is greater than $\epsilon$. Let's denote the enhanced density map as $\tilde{\mathbf{Y}}$. Next, we create a negative candidate map $\mathbf{NC}$, by subtracting the enhanced density map from the CB-map and then normalize the output, $\mathbf{NC} = \sigma(\mathbf{C} - \tilde{\mathbf{Y}})$, where $\sigma(\cdot)$ represents normalizing the input values to the range between $[0, 1]$. We can see that the purpose of the enhancement is that we would like to draw negatives that are far from the positives, the coefficient $\epsilon$ can control to what extent the CB-map is penalized when the positives are also near the centre.

Next we build a negative set with the same size as the positive set based on $\mathbf{NC}$. Values below zero in $\mathbf{NC}$ indicate the ground-truth density is higher than the CB-map, and we should not sample those locations as negatives. Then we apply a *Poisson* sampling process (Ghosh and Vogt, 2002) to create a negative point map, $\mathbf{NP} \in \mathcal{R}^{h \times w}$, based on $\mathbf{NC}$. Each location $(m, n)$ in $\mathbf{NC}$ is a candidate for $\mathbf{NP}$ to be selected with an inclusion probability, the value at location $(m, n)$ in $\mathbf{NP}$ is set to one if it is selected and otherwise zero, the inclusion probability for location $\mathbf{NP}(m, n)$ is $Pr(\mathbf{NP}(m, n) = 1) = \mathbf{NC}(m, n)$.

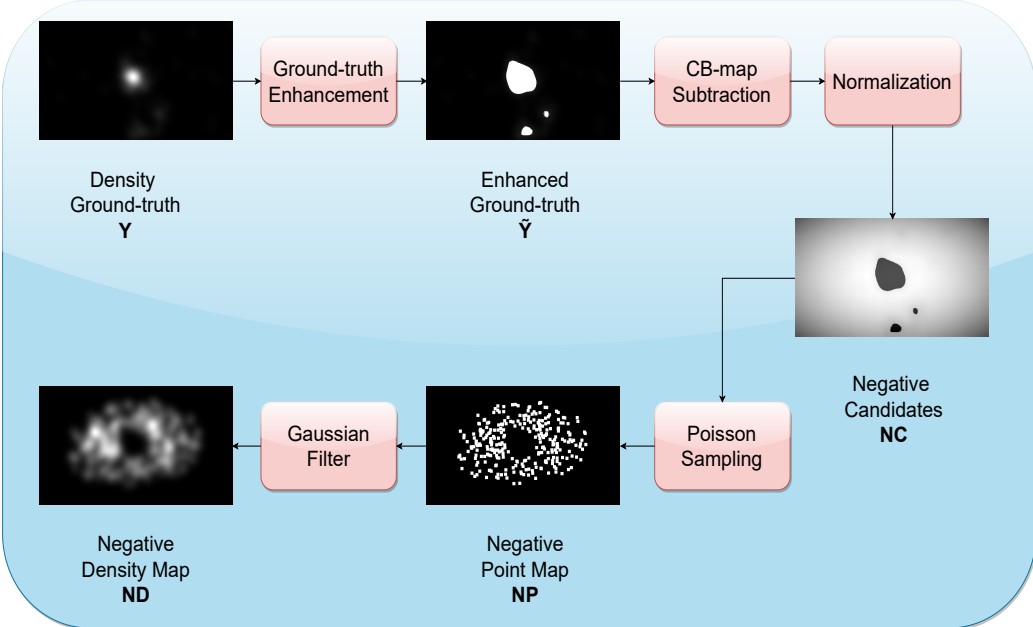

Figure 3: Flowchart of the proposed Centre-Neg solution.

The number of the negative points is set to the same number as the positive set as used in (Tatler et al., 2005a; Jia and Bruce, 2020). A negative density map is generated by applying a Gaussian filter on **NP**, denoted as **ND**. This process can be interpreted as the location is more likely to be sampled as a negative if its value is high in **C** and low in **Y**, which can penalize the CB-map without hurting the fixations, as shown in Fig. 4. Because the negative set of **NP** is created by sampling a subset of the CB-map, thus they are supposed to be highly overlapping, the score of CC between **ND** and **C** is 0.435. In contrast, the CC score between **ND** and **Y** is $-0.159$, which indicates the negative map is inversely correlated with the ground-truth; our negative map can avoid the region of fixations. Let's denote all non-zero elements in the map as a set, then we have $\mathcal{NP} \subseteq \mathcal{NC} \subseteq \mathcal{C}$. Meanwhile, the created negative set will not penalize fixations regardless of where they are, given $\mathcal{NP} \cap \mathcal{Y} = \oslash$ if $\epsilon$ is small enough.

Recall that both s-AUC and Centre-Sub cannot accurately evaluate saliency, because the most important central region is omitted due to the spatial bias. spROC and FN-AUC can only overcome the centre bias under a specific condition, their success still depends on the distribution of the fixations, as shown in Sec. 2.1. In contrast, Centre-Neg is proposed following a new idea that evaluates saliency without relying on spatial information (where the fixations are distributed), thus our method is more robust than the existing solutions. s-AUC and FN-AUC can be considered applying *Bernoulli* sampling on different populations (non-overlapped fixations from the same dataset and a subset of farthest neighbours respectively), while Centre-Neg applies *Poisson* sampling on the population of the processed CB-map.

Please note that all of the bias-specific metrics require to draw negative points, so they share the same complexity for sampling. The difference of complexity among the metrics

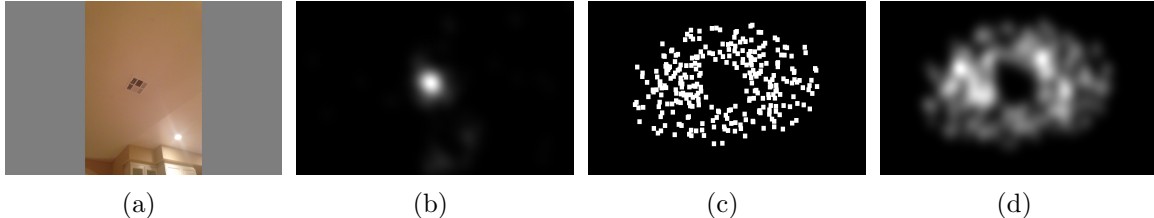

|        |        |        |        |
| :----: | :----: | :----: | :----: |
|  (a)   |  (b)   |  (c)   |  (d)   |

Figure 4: (a) Input image, **X**. (b) The density ground-truth, **Y**. (c) The negative point map, **NP**, created by Centre-Neg. (d) The negative density map, **ND**.

is how to build a negative candidate set. s-AUC, FN-AUC spROCand Centre-Sub need to iterate the whole dataset (assuming $N$ images) to build the candidate set, thus the complexity scales with the size of the dataet, $\mathcal{O}(n)$. While one main advantage of our Centre-Neg is that the candidate set $\mathcal{NC}$ does not rely on other images, so it will not rely on the size of the dataset, the time complexity of building $\mathcal{NC}$ is $\mathcal{O}(1)$. More importantly, our Centre-Neg can penalize the centre bias more effectively than the previous solutions. Our method differs from other bias-specific metrics in that the CB-map is utilized to draw negative samples, instead of mimicking the CB-map using fixations.

### 3.3. Metric-agnostic Combination

As studied in (Bylinskii et al., 2018), each saliency metric (including non bias-specific) may suffer from different biases in evaluation. Those bias-specific solutions could be impractical because most of them are computed based on AUC, which fails to penalize false positives with small values. (Any metric can be applied on the output of Centre-Sub, but the subtracted saliency map is not accurate.) To solve this problem, we apply the Centre-Neg in a metric-agnostic fashion. Let's apply the CC metrics as an example (as recommended by (Bylinskii et al., 2018)), but any saliency metric can be applied as a replacement.

Traditionally, we compute a saliency score between a prediction map **P** and its ground-truth **Y**. For instance, let's denote $CC^+$ as the CC score between a prediction and the ground-truth, $CC^+ = CC(\mathbf{P}, \mathbf{Y})\uparrow$, which is the widely used standard setting in evaluation, the higher the better. Most of the generic metrics (non bias-specific) only consider the correlation between predictions and ground-truth. Inspired by (Jia and Bruce, 2020), our solution also computes correlation between a prediction and its negative density map to penalize the CB-map, denoted as $CC^- = CC(\mathbf{P}, \mathbf{ND})$. Recall that **ND** is designed to have almost (controlled by $\epsilon$) no overlap with **Y**, thus the lower the score the better, $CC^- \downarrow$. Meanwhile, the CC score between the CB-map **C** and **ND** is expected to be high, given that the negative map is sampled from the CB-map, $CC(\mathbf{C}, \mathbf{ND})\uparrow$, a high CC score indicates the negative map can significantly penalize the CB-map.

Let's combine the two terms together to measure saliency predictions, $CC^* \uparrow = CC^+ \uparrow - CC^- \downarrow$. The first term is the commonly used component to measure saliency comparing the prediction and its ground-truth. The second term will be used for penalizing the CB-map. We can see that the combined term, $CC^*$ is still such that the higher the score the better, and vice versa for KLD.

Similarly, we can also apply the Centre-Neg on NSS, $\text{NSS}^* = \text{NSS}^+ - \text{NSS}^-$, where $\text{NSS}^+ = \text{NSS}(\mathbf{P}, \mathbf{YP})$ and $\text{NSS}^- = \text{NSS}(\mathbf{P}, \mathbf{NP})$, $\mathbf{YP}$ is the ground-truth point map. The combined $\text{NSS}^*$ is still such that the higher the score the better. Centre-Neg can also be applied on the metrics of the AUC family. Recall that all the AUC variants differ in how to sample a negative set, including s-AUC, FN-AUC, Judd-AUC and Borji-AUC (Judd et al., 2012). We can consider the negative point map $\mathbf{NP}$ as a set of negatives, which can be used to compute AUC scores, denoted as CN-AUC$^*$ in the experiment section. Although s-AUC, FN-AUC can also draw negative sets to combine with other metrics, we quantitatively show the proposed Centre-Neg can better penalize the CB-map, see Sec. 4.3.

## 4. Experiments

In this section, we mainly compare our proposed Centre-Neg against the other bias-specific metrics. Our experiment shows Centre-Neg can still effectively evaluate saliency in the scenario discussed in Section 2. Furthermore, we also compare different "early vision" saliency systems using our metric for completeness.

### 4.1. Datasets

In this study, four saliency datasets are applied to train or evaluate different models, including SALICON (Jiang et al., 2015), Toronto (Bruce and Tsotsos, 2005), MIT1003 (Judd et al., 2009) and CAT2000 (Borji and Itti, 2015). The SALICON dataset is a large-scale saliency dataset that contains $10,000$ images for training and $5,000$ images for validation. SALICON images are collected from the MS COCO image database (Lin et al., 2014). The MIT1003 contains $1,003$ images obtained randomly from Flickr creative (Russell et al., 2008). CAT2000 dataset includes $2,000$ images from the Internet using relevant keywords in different categories. Both of these datasets are used for the MIT/Tüebingen saliency benchmark (Kümmerer et al.). The Toronto dataset contains 120 natural images.

We only used the training set of SALICON to train a CNN model. SALICON dataset contains images with a resolution of $480 \times 640$. The validation set of SALICON and the other three datasets were used to measure and report the performance. The MIT1003 and CAT2000 images are of different sizes to the SALICON. We first zero-padded the images to the aspect ratio of $4:3$ and then resized them to match the size of the training set, as used in (Cornia et al., 2018; Cornia et al., 2016). For the Toronto dataset, we only resized the images for consistency.

### 4.2. Saliency Models and Measures

As a saliency prediction model, we re-purposed a vanilla ResNet-50 model (He et al., 2016) pre-trained on the ImageNet dataset. The backbone model captures visual features for object classification and is well suited for our application. We followed the multi-level strategy used in (Cornia et al., 2016; Kümmerer et al., 2014). The outputs of {$conv1$, $conv10$, $conv22$, $conv40$, $conv49$} layers are combined to predict the saliency map. We trained this model on the SALICON training set. The initial learning rate was set to 0.1 with a weight decay of $1e^{-4}$. The model trained for 10 epochs and the learning rate was reduced every three epochs by a factor of 0.1. The batch size was set to 8. The optimizer

was stochastic gradient descent, and the loss was set to mean squared error. We call this model, the CNN model in our experiments.

In addition to deep learning models, we also evaluate some traditional saliency systems that utilize low-level visual features: Itti (Itti et al., 1998), AIM (Bruce and Tsotsos, 2005), GBVS (Harel et al., 2006), SUN (Zhang et al., 2008), SDSR (Seo and Milanfar, 2009), CAS (Goferman et al., 2012), AWS (Garcia-Diaz et al., 2012), SWD (Duan et al., 2011) and ImageSig(RGB) (Hou et al., 2012). It has been shown that (Bruce et al., 2015; Borji et al., 2013; Zhang et al., 2008) those traditional methods could deliver different types of spatial biases due to the features they used. For instance, AWS is peripheral-biased while GBVS focuses more at the centre. Since, some of these approaches, e.g., CAS, are computationally expensive, we used the Toronto dataset to compare those models for efficiency.

### 4.3. Metric Comparison: Centre-Neg vs. Other Bias-specific Metrics

We first compare the proposed Centre-Neg with other bias-specific metrics, s-AUC, FN-AUC, spROC and Centre-Sub to show the advantage of our method. Please note that no saliency models are applied in this experiment, because the comparison among different metrics should be model-agnostic. As discussed in Sec. 2.1, the main drawback of Centre-Sub is that fixations near the centre of an image will be ignored, only peripheral regions are considered for evaluation, as shown in Fig. 5(the second row). For spROC, the number of fixations each annulus received could be very different. Fig. 5(the third row) shows the histogram plots for the examples, most of the fixations fall into the most central bins (first two columns), while the peripheral bins (last two columns) only receive few fixation points for evaluation. In this scenario, the AUC score achieved for each bin cannot robustly represent spatial biases. We apply the CB-map as a prediction using the partitioned bins from spROC, presumably there exists a strong central bias. However, examples in Fig. 5(the fourth row) show that the first column cannot obtain the highest AUC score. On the other hand, the AUC score obtained for the last bin could be as high as the first bin, which falsely indicates a consistent or peripheral preference of the model (CB-map in this case). The reason behind could be due to: a) the limited fixations (true positives) received in the peripheral region; b) AUC will not penalize false positives when their values are too small (Bylinskii et al., 2018).

Comparing with Centre-Sub, our Centre-Neg will not affect fixations (true positives) for evaluation, negative maps are added to penalize the centre bias, as shown in Fig. 5(the last row). Both s-AUC and FN-AUC can also draw negative maps for evaluation, we now further qualitatively compare the negative map created by Centre-Neg against the two metrics. From Fig. 6, the negative points of s-AUC (the second column) highly distribute near the fixation, thus the metric cannot accurately evaluate saliency. The negative map of FN-AUC is more directional than s-AUC, the fixation region is less affected. Our Centre-Neg can significantly avoid the fixation region by design, but still closely distributed near the centre region. This qualitative comparison shows our negative map can penalize the centre bias more accurately and efficiently than all of the other bias-specific metrics.

Following the metric evaluation used in FN-AUC, we can measure the quality of the created negative map by computing a CC score between the negative map (all denoted as **ND** in this case) and the CB-map, and a CC score between the saliency ground-truth **Y**

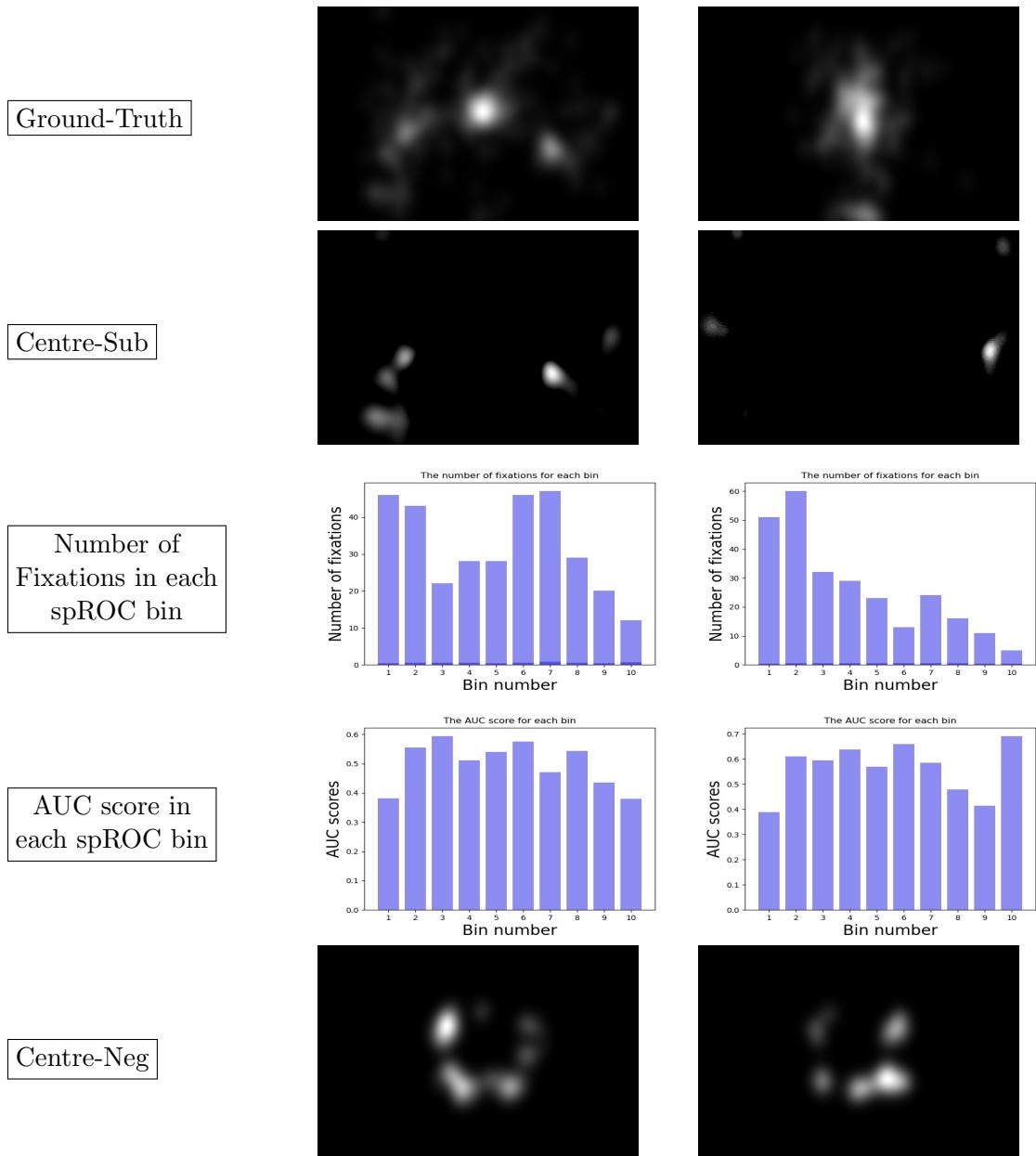

Figure 5: From the first row to the last: 1) Ground-truth saliency map. 2) The subtracted saliency map built by Centre-Sub. 3) The number of fixations within each partitioned bin built by spROC. 4) The AUC score of the CB-map for each bin using spROC. 5) The negative map created by Centre-Neg.

and the CB-map $\mathbf{C}$. Thus, the quality of a negative map can be formulated as CC($\mathbf{C}$, $\mathbf{ND}$) - CC($\mathbf{Y}$, $\mathbf{ND}$). The first term, CC($\mathbf{ND}$, $\mathbf{C}$), shows to what extent the negative map correlates with the CB-map, the higher the better. The second term, CC($\mathbf{ND}$, $\mathbf{Y}$), shows how the negative map correlates with the ground-truth, the lower the better. The subtraction of the

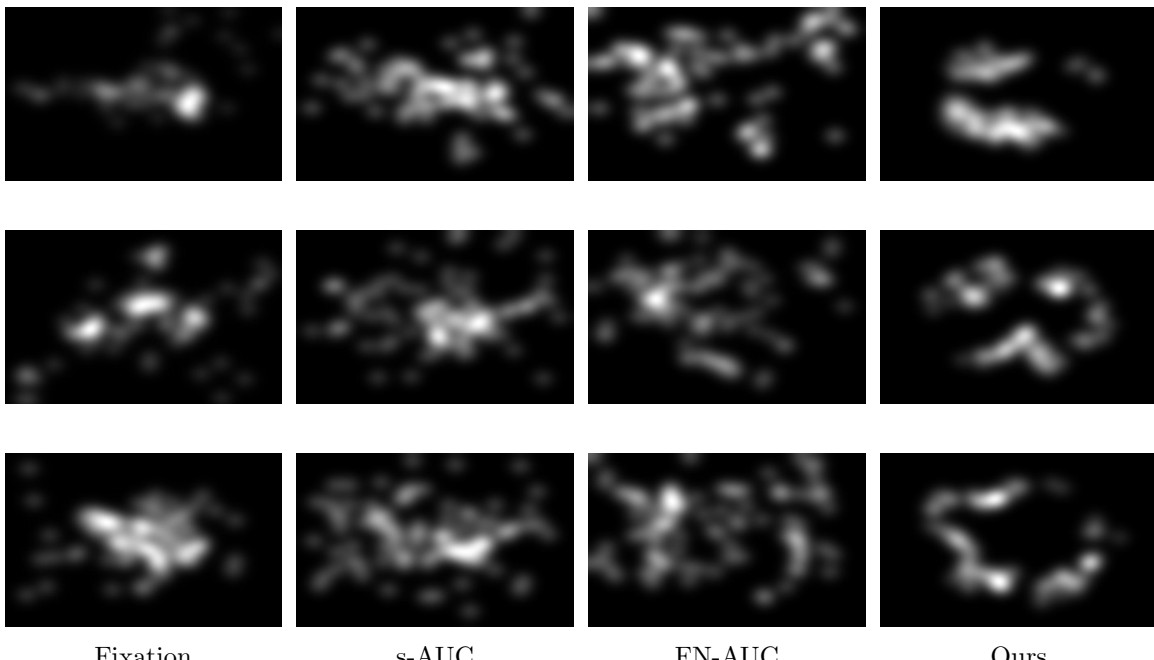

Fixation         s-AUC         FN-AUC         Ours

Figure 6: From left to right: (a) Fixation saliency map; (b) The negative map created by s-AUC; (c) The negative map created by FN-AUC; (d) The negative map created by Centre-Neg. Our method can create negative maps near the centre and avoid overlapping with the fixation region.

Table 1: Comparison of the quality of the negative maps drawn by s-AUC, FN-AUC and Centre-Neg.

| Method | Toronto | CAT2000 | MIT1003 | SALICON |
|---|---|---|---|---|
| s-AUC | 0.208 | 0.402 | 0.223 | 0.337 |
| FN-AUC | 0.452 | 0.189 | 0.368 | 0.534 |
| Centre-Neg | 0.661 | 0.724 | 0.448 | 0.717 |

two terms means the higher the score the better quality. Table 1 shows the scores achieved by applying the three methods on the four saliency datasets. *Not surprisingly*, our Centre-Neg achieves the highest scores on the four datasets. Please note that this quantitative comparison is simply shown for completeness. The negative map of our method **ND** is sampled based on the CB-map **C**, so it can obtain a higher correlation. However, this comparison is not completely trivial, because the negative maps of the other metrics are built based on all of the fixations. And the hidden assumption behind is that the map of all fixations (denoted as **A**) is supposed to approach (mimic) the CB-map, $\mathbf{A} \to \mathbf{C}$. Thus, the negative maps of all the metrics are assumed to be sampled from the same distribution (centre region).

Table 2: Comparison of different saliency systems on the Toronto dataset. The best result is highlighted in bold and the worst result is shown in bold.

| Method | SIM$^+$ | SIM$^*$ | KLD$^+$ | KLD$^*$ | CC$^+$ | CC$^*$ | NSS$^+$ | NSS$*$ | Judd-AUC | Borji-AUC | s-AUC | FN-AUC | CN-AUC$^*$ |
|---|---|---|---|---|---|---|---|---|---|---|---|---|---|
| CB-map | 0.369 | -0.090 | 1.260 | 0.411 | 0.397 | -0.161 | 0.969 | 0.020 | 0.802 | 0.786 | 0.507 | 0.603 | 0.554 |
| Itti (Itti et al., 1998) | 0.371 | 0.156 | 1.291 | -0.776 | 0.270 | 0.293 | 0.820 | 0.894 | 0.693 | 0.677 | 0.634 | 0.702 | 0.697 |
| AIM (Bruce and Tsotsos, 2005) | 0.382 | 0.154 | 1.357 | -0.679 | 0.312 | 0.304 | 0.896 | 0.897 | 0.725 | 0.719 | 0.666 | 0.726 | 0.733 |
| GBVS (Harel et al., 2006) | 0.487 | 0.145 | 0.851 | -0.448 | 0.569 | 0.262 | 1.519 | 0.763 | 0.829 | 0.815 | 0.636 | 0.748 | 0.675 |
| SUN (Zhang et al., 2008) | 0.348 | 0.129 | 1.430 | -0.643 | 0.215 | 0.227 | 0.650 | 0.698 | 0.665 | 0.654 | 0.608 | 0.658 | 0.660 |
| SDSR (Seo and Milanfar, 2009) | 0.413 | 0.197 | 1.096 | **- 1.106** | 0.403 | 0.424 | 1.096 | 1.174 | 0.763 | 0.757 | 0.695 | 0.785 | 0.787 |
| CAS (Goferman et al., 2012) | 0.438 | 0.205 | 1.029 | -0.918 | 0.449 | 0.422 | 1.271 | 1.232 | 0.781 | 0.767 | 0.687 | 0.779 | 0.773 |
| AWS (Garcia-Diaz et al., 2012) | 0.431 | 0.211 | 1.019 | -0.977 | 0.466 | 0.476 | 1.341 | 1.399 | 0.787 | 0.779 | 0.706 | 0.789 | 0.797 |
| SWD (Duan et al., 2011) | 0.454 | 0.120 | 0.935 | -0.361 | 0.575 | 0.194 | 1.523 | 0.565 | 0.836 | **0.829** | 0.621 | 0.740 | 0.633 |
| ImageSig (Hou et al., 2012) | 0.415 | 0.193 | 1.087 | -0.873 | 0.396 | 0.399 | 1.085 | 1.122 | 0.762 | 0.750 | 0.672 | 0.755 | 0.766 |
| CNN | **0.576** | **0.231** | **0.620** | -0.865 | **0.691** | **0.627** | **1.929** | **1.856** | **0.855** | 0.828 | **0.716** | **0.817** | **0.819** |

## 4.4. Results: Revisiting "Early" Vision Models on Centre-Neg

As discussed in (Bruce et al., 2015), different "early" vision systems utilize different visual features, some models could be centre-biased and some are peripheral-biased. To re-evaluate those low-level models using our solution, we report their results on the Toronto dataset in Table 2. The CB-map from the MIT/Tubingen benchmark (Judd et al., 2012) is also applied for comparison. Moreover, we also apply the metrics in both forms, the original form without negatives, and the form with the negative map, as discussed in Section 3.3.

The best result on each metric, with and w/o Centre-Neg, is highlighted in bold. Recall that a model would be less centre biased if $CC^+$ is lower than $CC^*$, the prediction is inversely correlated with the negative (centre) map, like SUN ($CC^+$:0.215) $\rightarrow$ ($CC^*$:0.227), and vice versa for GBVS and SWD. As discussed in (Bylinskii et al., 2018), the saliency metrics have different drawbacks, e.g., SIM is sensitive to the parameter of the Gaussian blur, and it penalizes more on false negatives over false positives. Thus, combining Centre-Neg with SIM also inherits those drawbacks and leads to a biased evaluation, i.e., SUN achieves lower $SIM^*$ score than $SIM^+$.

The CNN model achieves the best result on almost all of the metrics comparing to the other "early" vision systems. More importantly, Centre-Neg can penalize the CB-map significantly so that the CB-map achieves the worst results on all of the metrics, the worst results are highlighted in light grey. Interestingly, SUN (Zhang et al., 2008) achieves worse results than the CB-map when centre bias is not penalized, but SUN can outperform the CB-map when the centre bias is considered, the metrics combined with Centre-Neg. Likewise, Itti (Itti et al., 1998) and AIM (Bruce and Tsotsos, 2005) achieve lower scores than the CB-map on $CC^+$ and $NSS^+$, while they both outperform the CB-map after penalizing the centre bias. Considering spatial biases, AWS (Garcia-Diaz et al., 2012) and SDSR (Seo and Milanfar, 2009) have been proven to be less centre-biased (Bruce et al., 2015; Wloka and Tstotsos, 2016). Our experiment also validates this result, the CC scores of the two methods increase after applying Centre-Neg, which indicates the prediction could be inversely correlated with the negative map, $CC^- < 0$, the models are less biased towards the centre. In contrast, the most biased model, GBVS, is penalized by our Centre-Neg, the CC score drops significantly due to its strong centre bias, $CC^+ = 0.569 \rightarrow CC^* = 0.262$. Our result also shows SWD (Duan et al., 2011) also has a strong centre bias. And the proposed Centre-Neg can detect spatial biases encoded by each saliency model.

We refer the interested readers to study Bylinskii et al. (2018) for a detailed comparison of those saliency metrics, this experiment mainly shows our created negative map can be

combined with different metrics for the spatial bias issue. Further, we believe that a new metric should be proposed by *theoretically* showing the advantages over previous efforts; our Centre-Neg is proven to be more effective in Sections 3 and 4.3. The *empirical* results achieved in Table 2 are only shown for completeness.

## 5. Conclusion

In this paper, we have shown that all the previously proposed bias-specific metrics suffer from different drawbacks. This is especially true when most fixations are densely distributed near the centre; all the metrics will fail to measure saliency correctly. To solve this problem, we discussed the importance of position information in modelling saliency signal. We showed that the position information could be important for saliency when repeated objects are shown or there is no objects. Our proposed solution can build negative sets more effective, our results have quantitatively and qualitatively proven that Centre-Neg can better penalize the centre bias without affecting the ground-truth drastically. Further, our proposed Centre-Neg is efficient yet effective to apply, with a time complexity of $\mathcal{O}(1)$.

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
