# OpenReview forum: "Centre-Negative: An Effective and Efficient Solution to Centre Bias in Visual Saliency Evaluation"
_NeurIPS.cc/2023/Workshop/Gaze_Meets_ML — Submitted to Gaze Meets ML 2023_

### Official Review · Reviewer_yig5 · 2023-10-12
**Centre-Negative: An Effective and Efficient Solution to Centre Bias in Visual Saliency Evaluation**

**Rating:** 6
**Confidence:** 5

**Review:**

The authors claim to propose a novel region-based solution to mitigate centre bias during visual saliency evaluation. The manuscript is well-written, however, some issues warrant further clarification.
•	Can the authors provide a more detailed rationale for the need for a new bias-specific saliency metric? How do existing methods fail comprehensively? Also, the authors can comprehensively discuss some real-world medical-imagery-based computer vision applications that are affected by the centre bias in visual saliency detection.
•	What are the potential broader impacts of addressing centre bias in saliency detection, especially in the medical computer vision domain? Are there any known limitations or drawbacks to the Centre-Negative approach?
•	How did the authors identify that the datasets used in this study are the best representative of typical visual saliency scenarios?
•	How is ground truth determined for these datasets, especially concerning central biases?
•	Were any datasets with non-central biases used to validate the generality of the Centre-Negative approach?
•	Were any statistical tests conducted to determine the significance of the performance improvements claimed by the Centre-Negative approach over other solutions? How were these tests chosen and were multiple comparison corrections applied?
•	Are there any scenarios where Centre-Negative might not be the preferred solution?
•	How well does Centre-Negative generalize to real-world data, especially those with diverse visual contexts and varying central biases?
•	Were any real-world case studies conducted to showcase the efficacy of the Centre-Negative approach?
•	While the paper discusses a new metric, were any deep learning architectures optimized using this new metric? If model optimization was performed, how were the hyperparameters of the model selected and tuned?
•	How exactly does the Centre-Negative method handle dense fixation distributions near the centre, from a technical perspective?
•	Are there any computational or memory overheads introduced when applying Centre-Negative to existing metrics?
•	The time complexity of O(1) is mentioned. Can the authors elaborate on this claim, especially in scenarios with large images or datasets?
•	How do the authors envision the future evolution of bias-specific metrics in visual saliency detection? Are there any plans to extend Centre-Negative to other types of biases or applications?

---

### Official Review · Reviewer_QzkG · 2023-10-23
**Proposed a low complexity negative map construction algorithm and uses the negative map to calculate the saliency score, so as to reduce the spatial bias from the earlier saliency detection.Proposed a low complexity negative map construction algorithm and uses the negative map to calculate the saliency score, so as to reduce the spatial bias from the earlier saliency detection.**

**Rating:** 7
**Confidence:** 3

**Review:**

In this paper, the authors proposed a low complexity negative map construction algorithm which is called Centre-Negative algorithm. This method has low computation complexity of O(1). Centre-negative is used to construct the negative map to calculate the saliency score. Quality of the negative maps were compared among s-AUC, FN-AUC and Centre-Neg on four publically available saliency datasets and negative maps constructed with Centre-Negative show the highest score (which means the best quality).

Overall, this paper is well written. The methodology is clear and the experimental design  seems to be sound. The authors used extensive experiments show that the saliency metrics using the negative map constructed from the proposed Centre-negative algorithm can reduce the spacial bias.

---

### Official Review · Reviewer_BVum · 2023-10-24
**New gaze/saliency comparison metric proposed, but unfair characterization of competing metrics**

**Rating:** 5
**Confidence:** 4

**Review:**

The paper proposes a new metric for comparing predictions from saliency methods with ground truths collected with an eye tracker. This comparison is usually complicated because most of the gaze of a person tends to concentrate on the middle of an image. Several methods for dealing with this center bias have been proposed. The most successful ones might give much more importance to correct saliency predictions in the peripheral region of the image. The new proposed metric tries to balance the importance of central and peripheral regions when scoring saliency predictions while still penalizing predictions that just repeat the center bias. The main difference from other metrics is how negative maps are sampled to penalize predictions that are too similar to the center bias. In this case, only areas of the center bias that do not have a high value (above a threshold) in the ground truth saliency heatmap are penalized.

Strengths
- Good definition of the quality of a metric: The negative map of the proposed metric matches a decent definition of the quality of a negative map, CC(center bias, negative map) -CC(ground truth, negative map), CC is cross correlation. This quality definition says that the negative map (areas where predictions should be penalized when indicating saliency) should be different from the ground truth, i.e., the ground truth should not be penalized and be able to get a high score. Models predicting the center bias should be penalized, so the negative map should be similar to the center bias.
- Good experimental/quantitative comparison between several metrics: several metrics from the literature are included in their experiments and there is a fair discussion of differences between metrics in section 4.4, highlighting what region of the image each metric gives higher weight when calculating its score value.

Weaknesses
- Lack of discussions of problems of the proposed metric: according to the paper, the proposed metric seems to be better than the other metrics in every way.  One of its aspects that might be worse than other metrics, for example, is the presence of two additional parameters (threshold that chooses what a high value of the ground truth is, and standard deviation of the center bias Gaussian), which makes the metric potentially more variable than other metrics. The paper also does not indicate how they chose the threshold parameter or what the center bias Gaussian standard deviation is. It also does not analyze how much their metric results changes in regards to the choice of parameter.
- Not only definition of the quality of a metric: the definition of the quality of the metric probably inspired the proposed metric, or vice-versa, since both are very similar. So, it is not surprising that the proposed metric gets the best quality according to their definition. However, there might be other definitions of the quality of the metric that might correlate better with other of the baseline metrics. For example, depending on the downstream task, peripheral regions might be more important than the center regions, which would make the sAUC or other of the proposed metrics more suitable for a model comparison. Furthermore, contrary to what is said in the paper, the sAUC metric does not really omit center region: it only gives less weight to the center region in its score value.
- Not the only metric-agnostic negative map: different than what is claimed in the paper, other center bias techniques can be easily adapted to several metrics. For example, “A Locally Weighted Fixation Density-Based Metric for Assessing the Quality of Visual Saliency Predictions”, by Milind S. Gide and Lina J. Karam has the equivalent of the sAUC negative map applied to the NSS metric.
- Not the only metric that can have a low computational complexity: other methods of calculating a negative map could also use a spatial Gaussian as center bias. The negative map of the sAUC metric can be trivially changed to the centered spatial Gaussian without big changes to the final results of the metric in most cases. Furthermore, the use of a centered Gaussian as a center bias might not be suitable for all types of images. For example, in a chest x-ray dataset, there might be a high concentration of gaze in the apices and bases of the lungs, in addition to the central heart region. Also, since the metric is probably going to be calculated for the whole dataset, the complexity of the use of this metric in a dataset will be O(n), while the complexity of applying the sAUC metric over the dataset can also be O(n) (first iterate through the dataset to store the fixations such that they can be accessed at O(1), then do another pass through the dataset to calculate the metric for each of the saliency maps).
- Method not fully specified: the standard deviation of employed Gaussian center bias was not specified

Smaller problems
- Irrelevant sampling method comparison: the paper describes a difference between their method and other methods as using Poisson sampling vs Bernoulli sampling. This comparison is similar to the region-based vs fixation-based difference, but the paper makes it seems like it is a different thing. It is also possible to do a very simple conversion from Bernoulli sampling to Poisson sampling to any method, by simply representing fixations spatially in the image space. In other words, I do not believe this comparison is relevant.
- Extensive writing: the paper is probably longer than it has to be, leading to an unnecessary heavy reading. For example: the paper discusses negative properties from other metrics in sections 1, 2.1, 3.2, and 4.3; section 3.1 feels unnecessary

The proposed metric might be useful in the comparison of saliency predictions and eye tracker ground truths. However, the paper fails in making a fair discussion of the positives and negatives of all the metrics, highlighting problems in other metrics that are not really a problem, and making claims of contributions that are trivial changes to previous metrics.

---

### Meta-Review · Area_Chair_BDAW · 2023-10-26

**Recommendation:** Reject
**Confidence:** 3

**Metareview:**

This paper addresses the issue of spatial bias in visual saliency detection metrics and introduces a region-based metric-agnostic solution called Centre-Negative to overcome central bias problems. The proposed approach efficiently combines with existing metrics and outperforms other solutions by considering the importance of central regions in modeling saliency signals.

Several aspects of the work should be better discussed or improved. Discussion on the limitation of the current approach, for example, its sensibility to the hyperparameters. Discussion on the novelty of the current metric compared to the existing ones. Other existing central bias techniques should be discussed and/or included in the experiments. Furthermore, authors should better explain how ground truth is determined for the datasets used in the experiments, especially concerning central biases.

---

### Decision · Program_Chairs · 2023-10-26

Reject